# DNA methylation changes during long-term in vitro cell culture are caused by epigenetic drift

Julia Franzen [1], Theodoros Georgomanolis[2], Anton Selich[3], Chao-Chung Kuo [1], Reinhard Stöger [4], Lilija Brant[2], Melita Sara Mulabdić[1], Eduardo Fernandez-Rebollo[1], Clara Grezella[1], Alina Ostrowska[1], Matthias Begemann[5], Miloš Nikolić[1], Björn Rath[6], Anthony D. Ho[7], Michael Rothe[3], Axel Schambach[3], Argyris Papantonis [2,8] & Wolfgang Wagner [1✉]

Culture expansion of primary cells evokes highly reproducible DNA methylation (DNAm) changes. We have identified CG dinucleotides (CpGs) that become continuously hyper- or hypomethylated during long-term culture of mesenchymal stem cells (MSCs) and other cell types. Bisulfite barcoded amplicon sequencing (BBA-seq) demonstrated that DNAm patterns of neighboring CpGs become more complex without evidence of continuous pattern development and without association to oligoclonal subpopulations. Circularized chromatin conformation capture (4C) revealed reproducible changes in nuclear organization between early and late passages, while there was no enriched interaction with other genomic regions that also harbor culture-associated DNAm changes. Chromatin immunoprecipitation of CTCF did not show significant differences during long-term culture of MSCs, however culture-associated hypermethylation was enriched at CTCF binding sites and hypomethylated CpGs were devoid of CTCF. Taken together, our results support the notion that DNAm changes during culture-expansion are not directly regulated by a targeted mechanism but rather resemble epigenetic drift.

[1] Helmholtz-Institute for Biomedical Engineering, RWTH Aachen University Medical School, Aachen, Germany. [2] Center for Molecular Medicine, University of Cologne, Cologne, Germany. [3] Institute of Experimental Hematology, Hannover Medical School, Hannover, Germany. [4] School of Biosciences, University of Nottingham, Sutton Bonington Campus, Loughborough, Leicestershire, UK. [5] Institute of Human Genetics, Medical Faculty, RWTH Aachen University, Aachen, Germany. [6] Department for Orthopedics, RWTH Aachen University Medical School, Aachen, Germany. [7] Internal Medicine Department of Hematology, Oncology and Rheumatology, Heidelberg University Medical Center, Heidelberg, Germany. [8] Institute of Pathology, University Medical Center Göttingen, Göttingen, Germany. ✉email: wwagner@ukaachen.de

Cell preparations are often expanded in vitro for many passages to achieve enough material for basic research or cellular therapy. However, culture expansion has a severe and continuous impact on the growth, morphology, gene expression, metabolomics, and function of primary cells, until they ultimately enter a state of replicative senescence[1,2]. This is of particular relevance for the more than 1000 registered clinical trials with mesenchymal stem/stromal cells (MSCs; www.clinicaltrials. gov), which usually use high cell doses (often about $10^6$ cells/kg body weight) and therefore necessitate excessive culture expansion[3,4]. Furthermore, long-term culture of cells is often required for basic research and this may hamper the reproducibility of results. Tracking the impact of long-term culture is therefore an important aspect for quality control of cell preparations.

Long-term culture is reflected by highly reproducible DNA methylation (DNAm) changes at specific sites in the genome[5,6]. We have previously demonstrated that DNAm levels at only six cytosine/guanine dinucleotides (CpG sites) can be used to estimate passage numbers and cumulative population doublings (cPD)[7,8]. It therefore provides a reliable biomarker to estimate the state of culture-associated modifications. In our previous studies, we referred to these epigenetic modifications as senescence-associated DNAm changes, albeit it is unclear if the culture-associated DNAm changes are linked to the state of cellular senescence—we therefore changed the terminology into culture-associated DNAm changes. Furthermore, it is so far unclear how DNAm patterns evolve during culture expansion and why they occur at specific genomic regions.

At first sight, the culture-associated DNAm changes seem to be related to the DNAm changes that are acquired during the aging of the organism[9]. In fact, there is some overlap with epigenetic clocks for aging, but the two processes can be clearly discerned and it is yet unclear how the DNAm changes are governed. It is generally anticipated that DNAm changes during development are regulated by epigenetic writers—such as the de novo methyltransferases DNMT3A and DNMT3B, or TET methylcytosine dioxygenases[10]. If an epigenetic writer is targeted to a specific site in the genome the neighboring CpGs will most likely also be modified. Alternatively, the DNAm changes that accumulate during culture expansion might not be directly regulated but rather reflect dysregulation, as suggested for epigenetic drift during aging of the organism[11–13]. A better understanding of how DNAm changes evolve in culture expansion might shed light into the underlying process.

Furthermore, it is generally thought that DNAm patterns are identical on the forward and the reverse DNA strand (Watson and Crick strands)[14]. Using hairpin-bisulfite PCR[15] several studies have demonstrated that, despite the general preference for concordant DNAm on both strands, certain sites are specifically methylated on only one strand[16–18]. Such hemimethylation can be stably inherited over several passages and has been associated with CCCTC-binding factor (CTCF)/cohesin binding sites[19]. CTCF has been linked with changes in chromatin conformation and cellular senescence[20,21]. We have demonstrated that senescence entry upon extensive culture expansion is associated with a reorganization of CTCF into large senescence-induced CTCF clusters (SICCs)[20]. However, it remains unclear if and how the binding of CTCF changes during long-term culture.

In this study, we further investigated if culture-associated DNAm changes are caused by a targeted regulatory mechanism or rather by epigenetic drift.

## Results

### DNAm changes to track the process of culture expansion. We have previously identified DNAm changes during culture

expansion of mesenchymal stromal cells (MSCs) based on 27k Illumina BeadChip datasets, and thereby established a 6 CpG predictor to estimate passage numbers specifically for MSCs[7]. In continuation of this work, we utilized the meanwhile available 450k Illumina BeadChip datasets, which interrogate ~16 times more CpGs than the 27k version of the chip, and we also considered datasets of additional types of primary cells. We compiled 63 DNAm profiles of human primary MSCs ($n = 45$), fibroblasts ($n = 5$), and human umbilical vein endothelial cells ($n = 13$) with precise information on passage numbers (Supplemental Table S1). To identify individual candidate CpGs that become continuously hyper- or hypomethylated during culture expansion of these primary cell types, we filtered CpGs by Pearson correlation with passage number with an initial cutoff of $R > 0.7$ or $R < -0.7$: 646 and 2,442 CpGs passed these criteria, respectively (Supplemental Data 1). To further refine the list of candidates, we used more stringent filter criteria ($R > 0.8$ and $R < -0.8$; and a linear regression slope $m > 0.02$) to select 15 hypermethylated and 15 hypomethylated sites (Supplemental Table S2).

To develop a simplified and easily applicable biomarker for estimation of passage numbers based on targeted DNAm analysis with pyrosequencing, we then focused on two hyper- and two hypomethylated CpGs that cooperated best for prediction of passage numbers in the microarray training dataset. The four chosen CpGs were related to the genes *Arachidonate 12-Lipoxygenase* (*ALOX12*, cg03762994), *Docking Protein 6* (*DOK6*, cg25968937), *Leukotriene C4 Synthase* (*LTC4S*, cg26683398), and *TNNI3 Interacting Kinase* (*TNNI3K*, cg05264232; Fig. 1a). The long-term culture-associated DNAm changes at these CpGs were then tested and validated by pyrosequencing in MSCs, fibroblasts, and HUVECs at various passages ($n = 44$). Samples of the training set were all cultured until growth arrest and senescence was checked by staining with senescence-associated β-galactosidase (Supplemental Fig. S1a, b). Based on pyrosequencing results, we generated a multivariable model for epigenetic predictions of passage number ($R^2 = 0.81$; Fig. 1b and Supplemental Fig. S1c). Ten times 10-fold cross-validation of the pyrosequencing training dataset resulted in a $R^2 = 0.84$ and a root mean squared error (RMSE) of 3.9 passages. Subsequently, our epigenetic predictor for long-term culture was validated on an additional independent set of samples ($n = 83$; $R^2 = 0.74$; Fig. 1c and Supplemental Fig. S1d) by pyrosequencing. Thus, DNAm analysis at these four CpGs facilitates relative precise estimation of passage numbers and was applicable for different cell types.

It is conceivable, that culture expansion particularly impacts 5-hydroxymethylcytosine (5hmC) levels, which cannot be distinguished from 5-methylcytosine (5mC) by bisulfite treatment. To address this question, we used the TrueMethyl Array kit on three MSC donors in early (passage 4) and late (passage 10) passages to specifically look at the percentage of 5hmC at our 646 hyper- and 2442 hypomethylated culture-associated CpGs. Overall, we did not detect high levels of hydroxymethylation at early or late passages (mean estimated 5hmC levels about 0.3%) and neither hyper- nor hypomethylated culture-associated CpGs showed higher mean levels of 5hmC than other CpGs (Supplemental Fig. S1e).

**Hypomethylation during long-term culture is reversed as an early event during cell reprogramming into induced pluripotent stem cells (iPSCs).** Culture-associated DNAm changes are reversed in fully reprogrammed iPSCs[22,23], but the kinetics of this epigenetic rejuvenation have not yet been addressed and it was unclear if this occurs simultaneously with re-setting of age-associated DNAm[24,25]. We therefore utilized publicly available DNAm profiles of TRA-1-60 positive cells at various time points after retroviral reprogramming of fibroblasts with *OCT3/4*, *SOX2*,

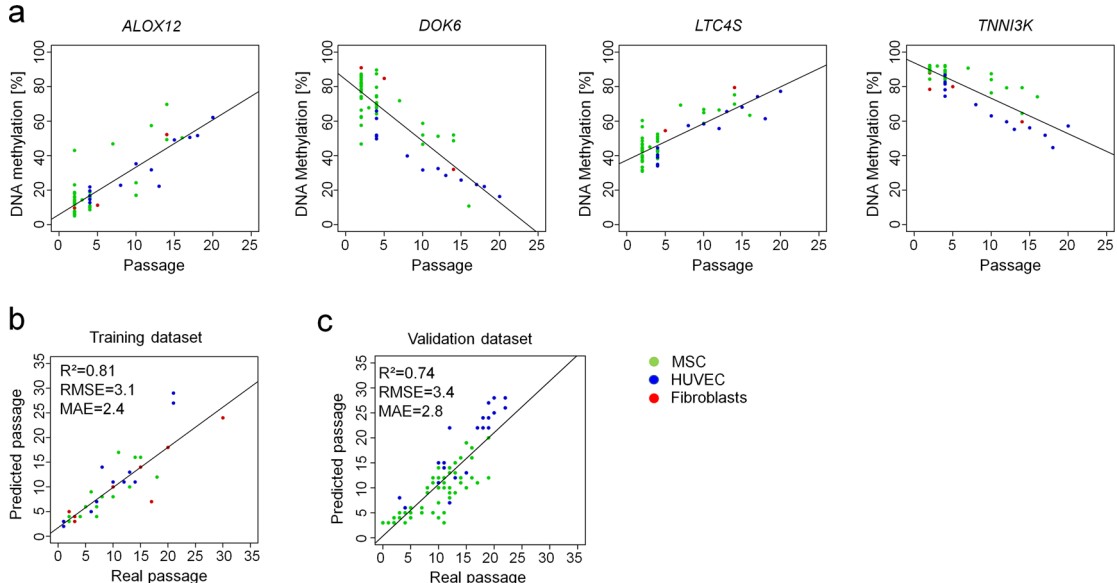

**Fig. 1 DNA methylation changes during long-term culture. a** Four CpGs were selected that revealed continuous changes during culture expansion in DNAm profiles of MSCs, fibroblasts, and HUVECs (all 450k BeadChip data, Supplemental Table S1). **b** DNAm at these four CpGs was then analyzed in a training set ($n = 44$, Supplemental Table S4) by pyrosequencing and the results were used to train a multivariable model to estimate passage numbers (RMSE root mean squared error, MAE mean average error). **c** This predictor was validated on pyrosequencing results of an independent set of samples ($n = 83$, Supplemental Data 2).

KLF4, and c-MYC[26]. Particularly CpGs that are hypomethylated at later passages become re-methylated between day 15 and day 20 after reprogramming, whereas hypermethylated sites are not consistently demethylated and sometimes gain methylation during reprogramming (Fig. 2a and b, Supplemental Fig. S2a). Notably, these epigenetic changes occur in parallel to the epigenetic modifications at pluripotency-associated CpGs (Supplemental Fig. S2b)[27]. We utilized the four new and five former CpG sites of our epigenetic signatures, which were represented by the 450k Illumina Microarray (Supplemental Fig. S2a) for predictions of passage numbers. While these estimates increased during culture expansion of MSCs, they declined around day 20 after reprogramming (Fig. 2c).

Importantly, culture-associated DNAm changes are distinct from age-associated DNAm changes, which correlate with chronological age rather than with passage numbers[9]. Age-related epigenetic signatures[25] are overall reset at the same time course (Supplemental Fig. S2c), a finding which is in line with another recent study[28]. In fact, the DNAm changes related to pluripotency, culture expansion, and aging follow the same kinetics (Fig. 2d and e) and the Pearson correlation of these DNAm changes is highly significant ($p < 2.2 \times 10^{-16}$).

During re-differentiation of iPSCs towards MSCs (iMSCs) there is an inverse switch in epigenetic patterns of pluripotency and culture expansion around day 7. While the long-term culture-associated DNAm changes are then continuously acquired upon differentiation of iPSCs, this is not observed for age-associated signatures[23] (Fig. 2a, b). Conversely, estimation of passage numbers gradually increased upon differentiation of MSCs towards iMSCs[23] (Fig. 2c). Taken together, epigenetic rejuvenation occurs simultaneously at aging and culture-associated CpGs. In contrast, to aging-associated DNAm changes[23,29] the culture-associated epigenetic modifications are then gradually reacquired over multiple passages of iMSCs.

**DNAm patterns do not reflect MSC clonality.** MSC preparations are heterogeneous and there is evidence that individual

subclones become dominant at later passages[30], which might contribute to culture-associated DNAm changes. Therefore, we aimed for a better understanding how DNAm patterns at neighboring CpGs evolve during culture expansion and how this is affected by the clonal composition of MSCs. We anticipated that tracking of DNAm patterns over several passages would provide insights into the changing composition of subclones within MSC preparations. To address this question, we used samples from a previously published study:[30] Umbilical cord-derived MSCs from two donors were transduced with lentiviral vectors containing random barcodes and three different fluorescent proteins. Flow cytometry and deep sequencing demonstrated that the diversity of cellular subsets declines and that senescent passages became oligoclonal (Fig. 3a and Supplemental Fig. S3a, b). We used barcoded bisulfite amplicon sequencing (BBA-Seq) to investigate DNAm patterns at the four culture-associated CpGs identified above, as well as the six CpGs of our previous predictor for replicative senescence (associated with the genes CASR, CASP14, GRM7, KRTAP13.3, PRAMEF2, and SELP)[7]. The combined BBA-seq measurements were then used to predict passage numbers. Taking all passages into account, the epigenetic estimations correlated well with the number of passages ($R^2 = 0.87$, Fig. 3b).

In contrast to pyrosequencing, BBA-Seq facilitates analysis of the succession of methylated and non-methylated CpGs within individual reads. To investigate such DNAm patterns, we focused on those amplicons that comprised several neighboring CpGs on the BBA-Seq reads (GRM7, CASR, LTC4S, DOK6, and ALOX12)—the other amplicons hardly comprised neighboring CpGs and could therefore not be included into this analysis. DNAm patterns overall remained stable or became even more diverse during culture expansion, which is also reflected by a moderate increase of the Shannon index in most amplicons (Fig. 3c and Supplemental Fig S3b–d). Thus, the development of DNAm patterns during long-term culture seems to be independent of the oligoclonal composition of MSCs at later passages.

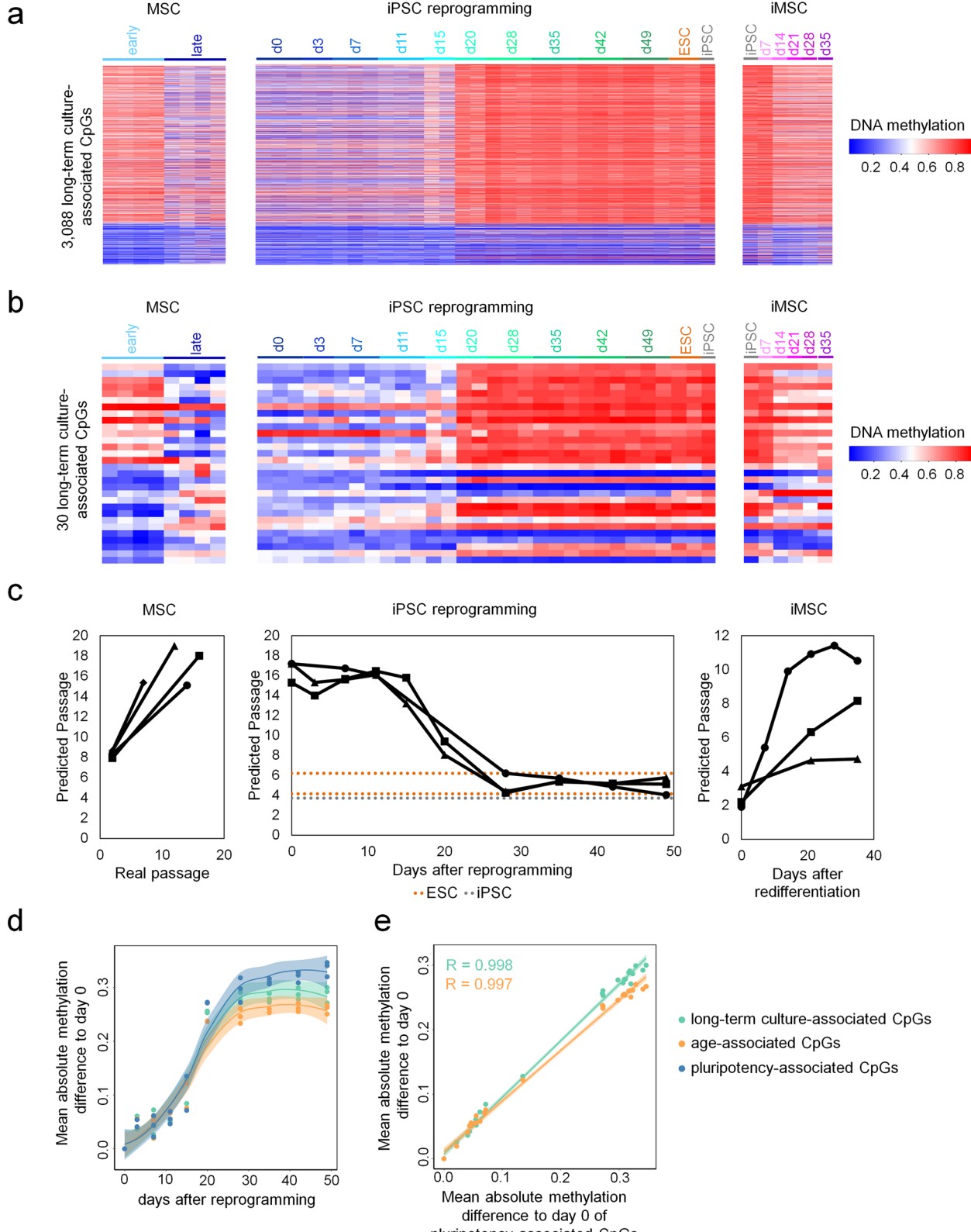

**Fig. 2 DNA methylation kinetics during reprogramming into iPSCs and re-differentiation to iMSCs. a**, **b** DNAm changes were analyzed in 3088 (**a**) or 30 (**b**) culture-expansion-associated CpG sites using DNAm profiles of MSCs of early (P2) and late (P7 to P16) passage (GSE37067[22]), in a dataset that analyzed DNAm changes at various time points during reprogramming of fibroblasts (GSE54848[26]), and during re-differentiation to iPSC-derived MSC (iMSC; GSE54767[23]). **c** Passage predictions of the three datasets described in (**a**) and (**b**). Passage predictions were calculated by using the mean of the predicted passages for the four new and five former CpG sites of our epigenetic signatures shown in Supplemental Fig. 2a. **d** and **e**) To further quantify the coincidence of pluripotency- with culture- or age-associated DNAm changes, we calculated the mean absolute methylation difference of all donors at each day of reprogramming to day zero. Culture-associated (3,088 CpGs), age-associated (99 CpGs), and pluripotency-associated CpGs (1432 CpGs) follow the same trend in methylation changes (**d**) and the differences of long-term culture- and age-associated sites are highly correlated to those of pluripotency-associated sites (**e**, Pearson correlation $R = 0.999$ and 0.997, respectively).

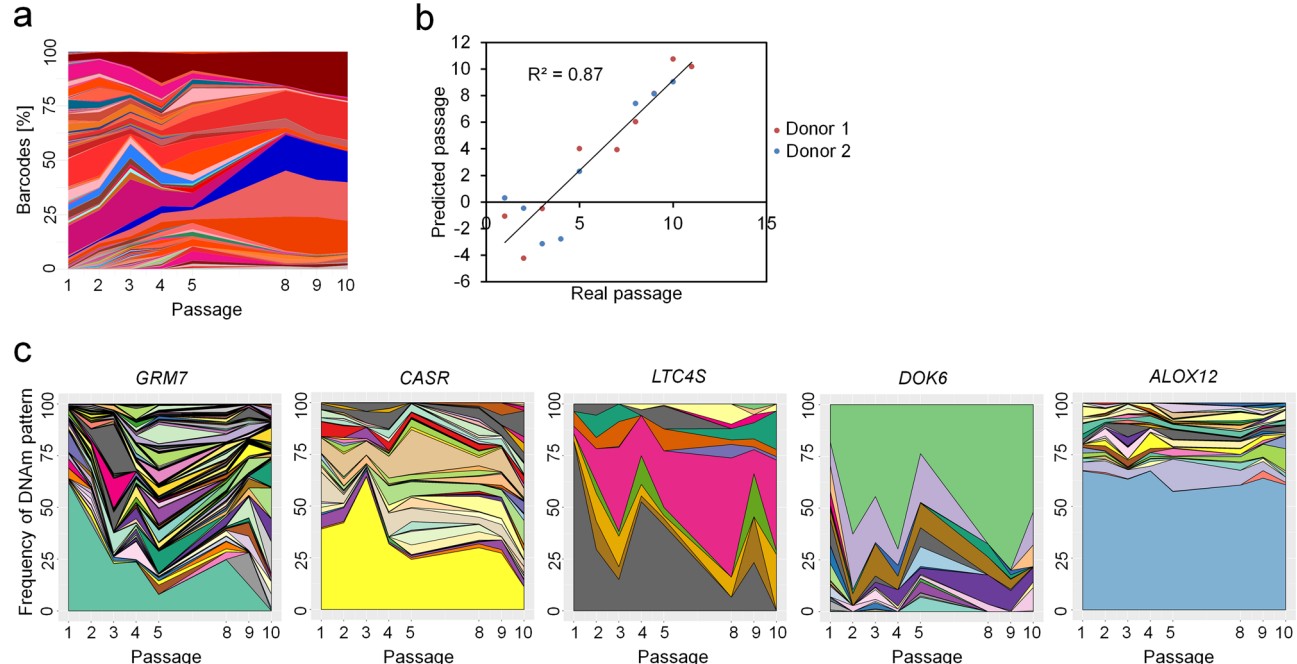

**Fig. 3 DNA methylation patterns are not affected by MSC clonality. a** Mesenchymal stem cells from umbilical cord were labeled with barcoded-RGB-vectors to correlate DNAm patterns with the composition of cellular subsets[30]. Deep sequencing analysis of the random barcodes demonstrates that the MSCs at passage 10 are oligoclonal. **b** These samples were subsequently used for DNAm analysis by barcoded bisulfite amplicon sequencing (BBA-Seq) at ten culture-associated CpGs. The predicted passage numbers based on these DNAm levels correlated with real passage numbers. **c** Changes in the frequency of different DNAm patterns over the passages are depicted for the neighboring CpGs within the amplicons of *GRM7* (22 CpGs), *CASR* (7 CpGs), *LTC4S* (4 CpGs), *DOK6* (7 CpGs), and *ALOX12* (9 CpGs). Each colored area corresponds to the frequency of one specific DNAm pattern, while the sum of all pattern frequencies adds up to 100% (Supplemental Data 3).

**Culture-associated DNAm patterns evolve stochastically at neighboring CpGs.** To gain further insight if culture-associated DNAm is regulated by targeted DNAm or rather by indirect epigenetic drift, we analyzed if DNAm is coherently modified at neighboring CpGs. If an epigenetic writer is targeted to a specific site in the genome the neighboring CpGs will most likely be coherently modified. To further investigate the dynamics of DNAm patterns we focused on the amplicon of *GRM7*, which comprised the highest number of CpG sites. The DNAm patterns fluctuated over subsequent passages (Supplemental Fig. S3d) and there was no evidence for continuous development of culture-associated modifications at this genomic region. In fact, the modifications seemed to be acquired randomly and there was hardly any correlation in DNAm between neighboring CpGs at a single read level (maximal Pearson correlation $R = 0.34$, Fig. 4a). On the other hand, several neighboring CpGs within the amplicon show also a very high correlation with passage numbers (Fig. 4b).

If the methylation changes at neighboring CpGs are acquired rather independently, then it should be possible to estimate probabilities for passage numbers also for the individual BBA-seq reads, based on the binary sequel of methylated and non-methylated CpGs[31]. To this end, we utilized the correlation with passage number at individual CpGs in the training set to establish a predictor based on single sequencing reads, as described in our previous work[31]. The algorithm estimates the likelihood for each pattern/read to belong to any passage between 0 and 50. By this approach, we clearly detect a high heterogeneity of sequenced DNAm patterns in samples of early and late passages. In tendency, samples at later passages comprised more BBA-seq reads that were predicted to correspond to higher passage numbers (Fig. 4c). To validate our findings we used BBA-Seq data of the *GRM7* amplicon from our previous study[8]. Notably, the

mean of single read predictions showed clear correlations between real and predicted passage numbers for training and validation set ($R^2 = 0.88$ and 0.72, respectively; Fig. 4d).

To address the question if DNAm patterns are identical on both complementary DNA strands we ligated hairpin oligonucleotides to connect the forward and reverse strands of individual DNA molecules (Fig. 4e)[15]. These hairpins also comprised a unique molecular identifier (UMI) in the loop region to adjust for potential PCR bias (Supplemental Fig. S4a). Eight out of the ten culture-expansion-associated regions encompassed suitable endonuclease restriction sites for targeted hairpin-ligation and could be further analyzed by BBA-Seq with primers specific for these hairpins (*CASR*, *GRM7*, *KRTAP13.3*, *PRAMEF2*, *SELP*, *DOK6*, *LTC4S*, *TNNI3K*). As a control, we considered an additional genomic region that was generally methylated (associated with the genes *C12orf12*). The accuracy of epigenetic predictions of passage numbers was similar when reads with the same UMI were only considered once (Supplemental Fig. S4b), indicating that potential PCR bias during amplification does not have a major impact on the mean DNAm levels. Subsequently we compared the DNAm patterns of the two complementary DNA strands. While we observed similar stochastic DNAm patterns as with conventional BBA-Seq, these patterns were overall faithfully shared between both DNA strands (Fig. 4f). Distinct CpGs of the long-term culture-associated sites exhibited slightly higher frequencies of hemimethylation than the frequencies observed at the control site (e.g., *DOK6*, *SELP*, *PRAMEF2*, *KRTAP13.3*, Supplemental Fig. S4c).

**Chromatin interactions at genomic regions with culture-associated DNAm changes.** Subsequently, we analyzed if genomic regions with gains or losses of DNAm might interact between different chromatin loops, which might be indicative for a

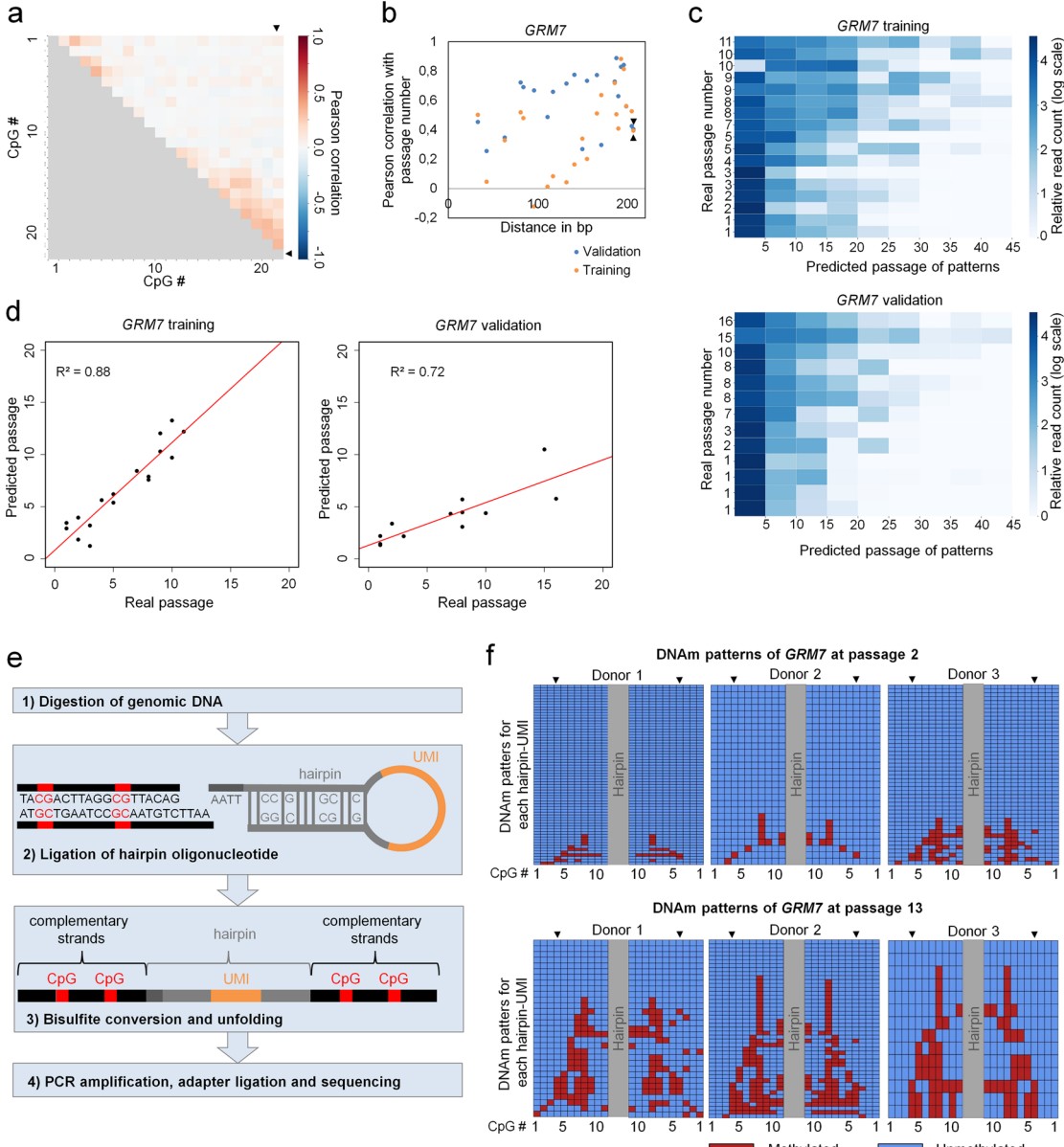

**Fig. 4 DNA methylation of neighboring CpG sites in the amplicon of *GRM7*. a** Pearson correlation of DNAm levels at neighboring CpG sites of the BBA-Seq amplicon of *GRM7*. The CpG site included in the Epigenetic-Culture-Expansion-Signature is CpG #22, depicted by black arrow heads. **b** Pearson correlation with passage number of neighboring CpGs in *GRM7*. The CpG site included in the Epigenetic-Culture-Expansion-Signature is the CpG at 207 bp, depicted by black arrow heads. Correlations are depicted for a training set (the two RGB labeled donors depicted in Fig. 3a and Supplemental Fig. 3a) and a validation set of our previous study[8]. **c** Heatmaps of single read predictions for the amplicon of *GRM7* show a high heterogeneity of predictions within each of the samples of the training and validation set. **d** The mean of the single read predictions correlated with real passage numbers. **e** Schematic presentation of DNAm analysis on complementary DNA strands using hairpin ligation and BBA-Seq (UMI = unique molecular identifier). **f** DNAm patterns of eleven neighboring CpGs in *GRM7* are depicted on complementary strands (MSC of three different donors in early and late passages). The CpG site included in the Epigenetic-Culture-Expansion-Signature is CpG #4, depicted by black arrow heads on both strands. Each row represents patterns for an UMI in the hairpin (Supplemental Data 4).

co-regulation. To this end, we exemplarily investigated chromatin interactions of four genomic regions with culture-associated DNAm changes (*ALOX12*, *LTC4S*, *CASR* and *KRTAP13.3*) using circular chromatin conformation capture (4C). Two independent MSC preparations at early (P2 and P3) and late (P7 and P9) passages revealed overall reproducible interaction profiles (Fig. 5a). For downstream analyses we only considered highly interacting regions that were categorized as nearbait (10 MB around the bait locus of interest) and as cis (all cis-contacts on the same chromosome), respectively. Although trans-chromosomal analysis showed a high number of reproducible interactions

(Fig. 5b) we excluded these sites from further analysis due to their high background signal, as commonly observed in such studies. The number of highly interacting sites called by *4Cker* in nearbait and cis remained similar with only a moderate increase between early and late passages (Supplemental Fig. S5a). Subsequently, we looked specifically for those interacting sites that showed reproducible differences between early and late passages in both donors (FDR adjusted *p*-value < 0.05): A similar number of interactions revealed such significant gains and losses during culture expansion (Supplemental Fig. S5b). Although this analysis was limited to the four exemplary regions with culture-associated DNAm

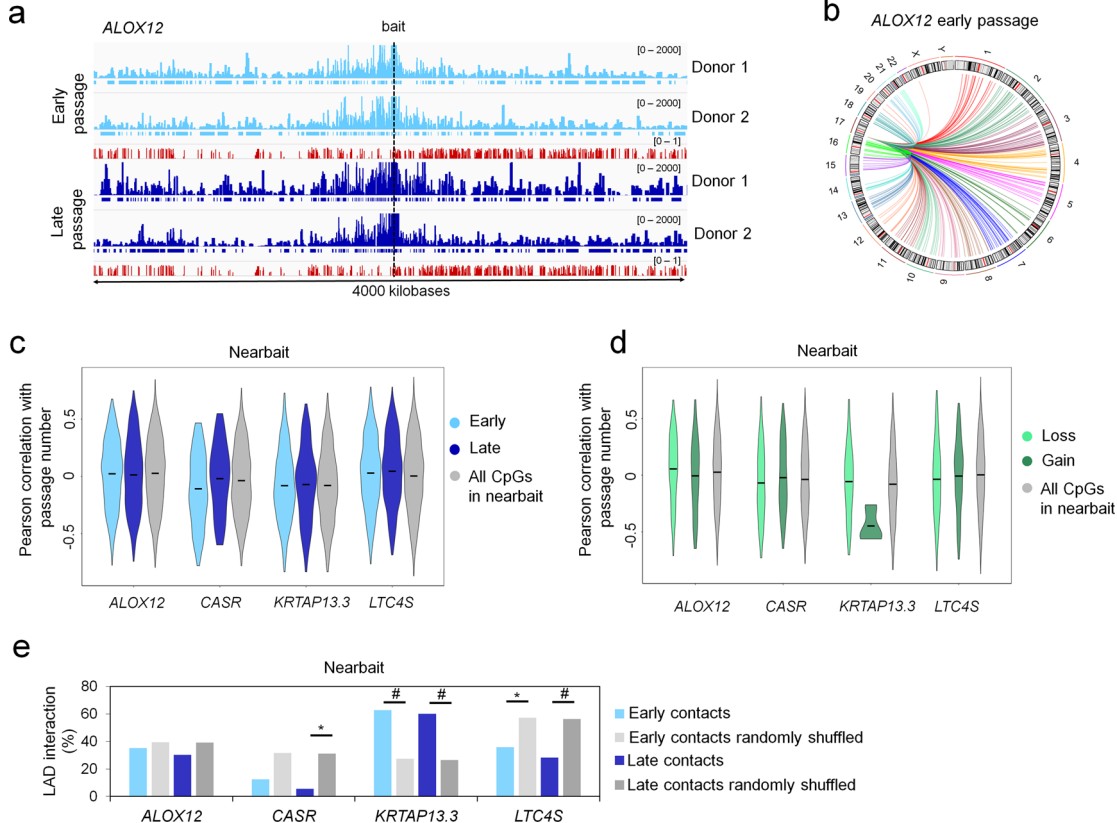

**Fig. 5 Circular chromatin conformation capture (4C) of culture-associated CpGs. a** Integrative Genomics Viewer (IGV) overview of the bait region of *ALOX12* (chr17:4741221-8741221; hg19) in MSCs of two donors at early (P2 and P3, light blue) and late (P7 and P9, dark blue) passage. Sequencing peaks are presented as normalized counts. High-interacting regions called by 4 Cker tool are indicated by horizontal bars beneath the peaks. Mean methylation levels of CpGs on the Illumina 450k BeadChip are depicted for MSCs at early (*n* = 5; P2-P3) and late passages (*n* = 5; P7-P13; GSE37067; red)[22]. **b** Circos-plot of highly interacting regions across different chromosomes (trans) is exemplarily depicted for *ALOX12* (interactions are reproducible in two MSC preparations at early passage). **c** For each CpG represented on the Illumina 450k Bead Chip the Pearson correlation between DNAm and passage numbers was calculated based on DNAm profiles of 63 DNAm profiles (Supplemental Table S1). Violin plots show the distribution of these culture-associated DNAm changes at CpGs in the nearbait regions of *ALOX12, CASR, KRTAP13.3*, and *LTC4S*. This was analyzed for 4C interactions in early passaged cells (light blue) or late passaged cells (dark blue) in comparison to all CpGs of the nearbait region (gray; means depicted by black bars). **d** In analogy to **c** the violin plots depict the Pearson correlation of DNAm with passage number of CpGs in the significantly differential contacts between early and late passage cells. **e** Enrichment of lamina-associated domains (LADs) of a publicly available dataset of human fibroblasts[32] within high-interacting sites (nearbait region) of the four baits compared to the mean interaction frequency of random background regions of the same size (shuffled along the nearbait region 1000 times). Significance was estimated by Fisher's exact test (*$p < 0.05$; **$p < 0.01$; ***$p < 0.001$; #$p < 0.0001$). Raw data of the 4C experiment was uploaded to Gene Expression Omnibus (GEO): GSE144196.

changes, the results indicate that there are reproducible changes in nuclear organization between early and late passages.

Next, we investigated if chromatin interactions preferentially occurred at other genomic regions that become methylated or demethylated during culture expansion. As mentioned above, we used the 63 DNAm datasets of cells at different passages (Supplemental Table S1) to determine the Pearson correlation between passage number and DNAm level for each CpG on the 450k Bead Chip. Overall the distribution of Pearson correlations was similar at CpGs within nearbait contacts for each of our four culture-expansion-associated genomic regions (at either early or late passage) as compared to all CpGs of the nearbait region (Fig. 5c). In analogy, the correlation of DNAm with passage number was similar at cis-contacts (Supplemental Fig. S5c). Alternatively, we focused on those genomic regions that revealed significant gains and losses in chromatin interactions between early and late passage: again the distribution of Pearson correlations was very similar with random CpGs at nearbait and cis-interacting regions (Fig. 5d and Supplemental Fig. S5d). Only for those nearbait interactions of *KRTAP13.3* that were

significantly gained at later passage we observed hypomethylation during culture expansion. In fact, the keratin-associated protein locus (*KRTAP*) was previously shown to have an exceptionally large differentially methylated region during culture expansion, which may explain this enrichment in the differentially methylated nearbait region[22]. However, this finding might also be coincidental since the significant gains of interaction at *KRTAP13.3* included only 6 CpGs of the Illumina Bead Chip. Overall, our results did not indicate that the four culture-associated genomic regions revealed clear enrichment of interaction with other culture-associated CpGs at nearbait or cis-contacts.

To analyze if the 4C interactions were related to lamina-associated domains (LADs) we used a publicly available dataset of fibroblasts[32]. We and others previously demonstrated that hypomethylation during long-term culture occurs preferentially at LADs[6,33,34]. This is supported by the observation that only the hypomethylated region of *KRTAP13.3* was enriched in chromatin interactions at LADs. The enrichment was determined by comparing the overlap of the 4C interactions with LADs in

comparison to the overlap of random regions (1000 times shuffled random regions with the same amount and size as the 4C interactions). In contrast, such overlap with LADs was rather depleted for hypermethylated regions (*ALOX12*, *CASR,* and *LTC4S*; Fig. 5e and Supplemental Fig. S5e). Taken together, our exploratory analysis did not indicate that genomic regions with culture-associated DNAm changes have enriched interaction with each other. It is therefore rather unlikely that the culture-associated CpGs are synchronously regulated at the interaction sites of different chromatin loops.

**DNAm changes during long-term culture are related to CTCF binding sites.** During culture expansion and upon entry into replicative senescence the cell nuclei become much larger and CTCF was shown to reorganize into large senescence-induced CTCF clusters (SICCs) in HUVEC and IMR90 cells[20]. When we analyzed the distribution of CTCF in MSCs by fluorescence microscopy we also observed increased co-localization of CTCF in the larger nuclei of senescent MSCs (Fig. 6a). We then asked the question if CTCF binding at specific genomic locations is also changed during culture expansion. To address this question, we performed chromatin immune precipitation (ChIP) with MSCs of early and late passage ($n = 3$). Overall, the ChIP seq peaks were in line with previous data of embryonic stem cell-derived MSCs[35] (Supplemental Fig. S6a) and they centered clearly around predicted CTCF binding motives (Fig. 6b). We were then looking for differential CTCF peaks between early and late passage. Transforming the data onto M (log ratio) and A (mean average) scales (MA-plot) revealed that there are no highly abundant differential peaks between early and late passages (Fig. 6c). Furthermore, Spearman correlation of normalized read counts provided further evidence that ChIP seq profiles of early and late passages are highly correlated (Supplemental Fig. S6b). Thus, CTCF binding appears to be relatively stable during culture expansion, despite dramatic increase in nuclear size, reproducible chromatin conformation changes, and reorganization of SICCs.

We then analyzed if chromatin interactions of the four culture-associated regions in our 4C data were related to CTCF sites. In fact, nearbait and cis-interacting regions of two hypermethylated sites (*ALOX12* and *LTC4S*) exhibited significant enrichment of binding motifs for CTCF and CCCTC-binding factor like (CTCFL; Supplemental Fig. S6c and d). Similar results were observed using our CTCF ChIP-seq data of MSCs (Supplemental Fig. S6e).

Subsequently, we analyzed if CpGs that become either hyper- or hypomethylated during culture expansion are related to CTCF binding sites. To provide enough genomic regions for such statistical analysis, we used our set of 646 hyper- and 2442 hypomethylated CpGs that were selected with the less stringent filter criteria as described above. In fact, hypermethylated CpGs were enriched at CTCF binding sites, whereas genomic regions that become hypomethylated were almost devoid of CTCF (Fig. 6d). This was consistent and highly significant ($p < 10^{-51}$) for all three donors (Fig. 6e). For comparison, we have also analyzed 2000 randomly chosen CpGs from the Illumina 450k BeadChip. These random sites also revealed a CTCF-ChIP-seq peak around the CpGs because CTCF-binding motifs usually comprise CpGs, resulting in a bias for these regions. However, our statistical analysis demonstrated, that the CTCF ChIP-seq signal at the random CpGs was significantly lower than at CpGs that become hypermethylated, and significantly higher than for CpGs that become hypomethylated during culture expansion (Fig. 6e).

## Discussion
The continuous and highly reproducible nature of culture-associated DNAm changes may suggest that this process is tightly

controlled[36]. Nevertheless, several of our results indicate that the process is rather associated with stochastic epigenetic drift, which does not involve site-specific targeting of regulatory protein complexes: (1) Resetting of culture-associated DNAm during reprogramming into iPSCs seems to occur synchronously with the epigenetic changes in pluripotency genes, which is directly linked to the epigenetic transition itself. Furthermore, it occurs simultaneously with resetting of age-associated DNAm, which has also been attributed to epigenetic drift[11–13]. (2) The culture-associated DNAm patterns do not reflect MSC clonality and become even more diverse in oligoclonal cell preparations at late passages. If these epigenetic modifications were evoked by targeted regulation in individual subclones specific patterns should become dominant, too. (3) BBA-seq analysis demonstrated that culture-associated DNAm patterns do not develop in an additive manner at neighboring CpGs and this was further supported by hairpin sequencing of the complementary DNA strands. If an epigenetic writer is targeted to a specific site in the genome the neighboring CpGs will most likely also be modified, as observed for CRISPR-guided approaches of epigenetic writers that coherently modify neighboring CpGs[37]. (4) There was no evidence that culture-associated DNAm is coherently modified at interacting chromatin domains. It is known that DNMTs accumulate in replication foci or punctate heterochromatic foci[38,39] and hence, it might be speculated that a targeted mechanism of an epigenetic writer would also involve the interaction of culture-associated chromatin domains. If culture-associated DNAm changes are not governed by targeting of epigenetic writers, there needs to be another—yet unknown— mechanism that disposes specific genomic regions to epigenetic drift.

Cell culture is evidently associated with major changes in chromatin structure. The nucleus becomes much larger at later passages, while chromatin volume decreases due to extensive reorganization of hetero- and euchromatin conformation[40,41]. Nuclear depletion of HMGB2 and its induction of CTCF clustering are early events on the path to replicative senescence, which disturb the chromosomal 3D organization[20]. Our 4C analysis supports the notion that the 3D chromatin structure undergoes highly reproducible changes during culture expansion, while these changes do not seem to be associated with changes in DNAm. It has been demonstrated that CTCF occupancy, which is to some extent cell-type specific, is also linked to differential DNAm[42]. Furthermore, some CTCF sites may function as a bifurcation point defining the differential methylation landscape[43]. In this regard, it was unexpected that we did not observe clear differences in the CTCF ChIP-seq data of early versus late passages. On the other hand, hypermethylation seems to occur preferentially at CTCF binding sites, whereas hypomethylation occurs apart from CTCF binding sites and preferentially at LADs. This might partly be attributed to the fact that the hypermethylated CpGs have rather low DNAm levels at early passage, which may support binding of CTCF to the corresponding motive at early passage[42,43].

Taken together, the results of this study support the notion that the given chromatin conformation favors site-specific epigenetic drift over subsequent passages—but they also provide room for additional hypotheses, which have not been addressed with our current study. We have previously demonstrated that some transcription factor binding motifs (e.g., EGR1, TFAP2A, and ETS1) were enriched in senescence-associated differentially methylated regions and in the promoters of differentially expressed genes[44]. Thus, the binding of these transcription factors might be affected by DNAm or vice versa[44]. It is also conceivable that DNAm changes during long-term culture are indirectly mediated by the histone code, insulators, chromatin loops, and the overarching nuclear structure. Furthermore, the culture-

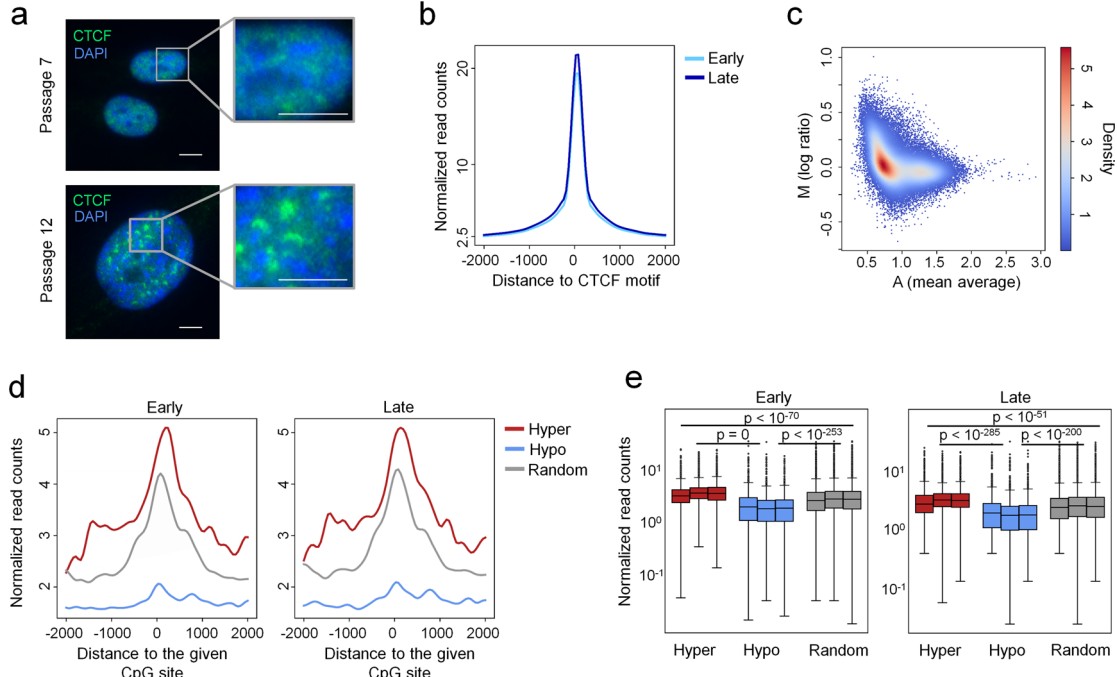

**Fig. 6 CTCF binding in cells of early and late passages. a** High-resolution microscopy pictures of MSCs at median passage (P7) and late passage (P12). Staining of cell nuclei with DAPI and CTCF reveals co-localization of CTCF particularly in the larger nuclei of late passage MSCs (size bars = 5 µm). **b** CTCF ChIP-seq signals of cells in early (P2) and late (P8–P14) passage are centered around predicted CTCF binding motifs. **c** MA plot of CTCF ChIP-seq signals of cells in early (P2) and late (P8–P14) passage. Mean average (A) is the mean of the normalized read counts of early and late passages, whereas log ratio (M) is the ratio of the normalized read counts of late passages over early passages. **d** Lineplots depict the CTCF ChIP-seq signals of cells in early (P2) and late (P8–P14) passage centered around hyper- or hypomethylated culture-associated CpGs or 2000 randomly picked CpGs of the Illumina 450k BeadChip. **e** Statistical testing of the ChIP-seq signal differences of three donors in early (P2) and late (P8–P14) passages at the CpG subsets shown in (**d**). CTCF ChIP-seq signals at hypermethylated CpGs are significantly enriched in comparison to randomly chosen CpGs, while CTCF ChIP-seq signals at hypomethylated sites are significantly depleted (Dunn's test on the quantile normalized reads of the 2000 bp window). Raw data of the CTCF ChIP-seq experiment was uploaded to Gene Expression Omnibus (GEO): GSE144196.

associated differentially methylated regions might be related to nucleosome size. However, for such analysis it needs to be considered that the Illumina Bead Chip data has the limitation that it does not address all CpGs of the genome. Furthermore, we have only exemplarily analyzed few amplicons with BBA-seq, hemimethylation analysis, and 4C. It is therefore unclear if the findings of our study are really representative of the entire genome. Importantly, the culture-associated DNAm changes might also be influenced by specific culture conditions or cell types and this deserves further analysis in the future.

Last but not least, the functional relevance of DNAm changes is still unclear. Overall, culture-associated CpGs are rather depleted from CpG islands and their shore/self-regions, while they are enriched in open sea, 3′UTR and intergenic regions[8]. Thus, a direct link to a specific gene promotor is often not possible. It has been shown that culture-associated DNAm is only partly associated with gene expression changes of the corresponding genes[22,45,46]. Yet, it would be an oversimplification to only consider transcriptional regulation as being functionally relevant. Chromatin conformation, loop structures, histone modifications, and LADs may favor modulation of DNAm at specific sites in the genome—and on the other hand, culture-associated DNAm may stabilize such chromatin features. In future research, different levels of chromatin organization should be considered to fully understand the underlying mechanism that drives epigenetic drift during long-term culture.

## Methods
**Identification of culture-associated CpGs**. We compiled 64 published and newly generated DNAm datasets of untreated primary cells with reliable information on

passage numbers (all Illumina 450k Methylation BeadChip; Supplemental Table S1). For further analysis we excluded CpGs from X and Y chromosomes and performed k-nearest neighbor imputation and quantile normalization using the R packages impute and lumi[47], respectively. An outlier test using the R package car was performed resulting in the exclusion of one sample (GSM1004625). Initially, we selected culture-associated CpGs based on Pearson correlation of DNAm levels (beta values) and passage numbers at a threshold of $R > 0.7$ or $R < -0.7$ (646 hyper- and 2442 hypomethylated CpGs, respectively; Supplemental Data 1). Subsequently, we used a more stringent threshold to identify the best candidates for targeted analysis: $R > 0.8$ or $R < -0.8$; and the slope of regression $m > 0.02$ (highest ranked 15 hyper- and 15 hypomethylated CpG sites; Supplemental Table S2). Using the R package leaps we calculated the best multivariable linear regression model including two hypo- and two hypermethylated CpG sites.

**Analysis of DNAm changes during reprogramming into iPSCs**. The kinetics of culture-associated DNAm changes during reprogramming of fibroblasts into iPSCs were investigated using the dataset of Ohnuki et al. (GSE54848; 450k BeadChip)[26]. For comparison we used DNAm profiles of MSCs at early and late passage (GSE37067)[22] as well as DNAm profiles during re-differentiation of iPSCs to iMSCs (GSE54767)[23]. We focused on three different sets of culture-associated CpGs: (1) 30 CpGs (filtered by $R > 0.8$ or $R < -0.8$, $m > 0.02$, described above); (2) 3088 CpGs (filtered by $R > 0.7$ or $R < -0.7$, described above); and (3) 9 CpGs of our epigenetic signatures (*KRTAP13.3, SELP, CASP14, TNNI3K, DOK6, CASR, GRM7, LTC4S, ALOX12*)[7]. Furthermore, we focused on 1432 pluripotency-associated CpGs[27] and 99 age-associated CpGs[25] of the Illumina 450k BeadChip. Heatmaps were produced with R package gplots, Pearson correlation, and corresponding p-values were calculated with R package stats.

**Cell culture**. All human cell samples were taken after informed and written consent was obtained from donors and the study was specifically approved by the ethics committees of RWTH Aachen University Medical School (permit numbers: EK300/13, EK163/07, EK 187/08), University of Heidelberg, and Hannover Medical School. MSCs were isolated from the bone marrow of donors undergoing orthopedic surgery ($n = 11$; BM-MSC RWTH Aachen)[48], from bone marrow aspirates of allogeneic hematopoietic stem cell donors ($n = 8$; BM-MSC University

of Heidelberg)[49], from subcutaneous adipose tissue lipoaspirates (n = 3, AT-MSC RWTH Aachen)[50] and from umbilical cord pieces (n = 2; UCP-MSC; University of Hannover)[30]. Fibroblasts were isolated from dermis (n = 4; RWTH Aachen)[7]. Human umbilical vein endothelial cells were isolated from umbilical cords of healthy donors after cesarean sections (HUVECS; n = 4; RWTH Aachen)[51] or obtained from Lonza (n = 3, Basel, Switzerland)[8]. All samples were taken after informed and written consent and the study was specifically approved by the ethics committees of the corresponding Universities. All cell preparations were thoroughly characterized (including morphology, immunophenotype, and three lineages in vitro differentiation potential) and culture conditions were used as described in detail in our previous work[7,8,30,48–50]. In addition, HUVECS were cultured on 0.1% gelatin in M199 medium (Thermo Fisher, Waltham, USA) supplemented with 20% fetal calf serum (FCS) (Gibco Thermo Fisher), 1% penicillin/streptomycin (PAA), 0.1% heparin (5000 IU/ml, Ratiopharm) and 50 µg/ml endothelial cell growth supplement (ECGS) (Sigma-Aldrich, St. Louis, USA). For long-term culture all cells were passaged at ~90% confluency and reseeded at 10,000 cells/cm². Beta-galactosidase (SA-ß-gal) staining was performed using the Senescence Detection Kit ab65351 (Abcam, Cambridge, UK). cPD were calculated as described before[50].

**Hydroxymethylation analysis.** Hydroxymethyltaion was analyzed with the TrueMethyl Array kit (Cambridge Epigenetix, Cambridge, UK) according to the manufacturer's instructions. In short, three MSC donors of early passage (passage 4) and late passage (passage 10) were each divided into two samples: one sample was oxidized and bisulfite converted (OxBS), the other sample was just bisulfite converted (BS). Oxidation prior to bisulfite treatment leads to the conversion of 5hmC to 5-formylcytosine (5fC), which is decarbonylated and deaminated to uracil upon bisulfite treatment, similar to non-methylated cytosines. Both OxBS and BS samples were finally investigated by Illumina 450k BeadChip microarrays (performed by Life & Brain, Bonn, Germany). Beta values of the microarrays were provided by Illumina's GenomeStudio software and hydroxymethylation was calculated as the difference of beta values of BS samples subtracted by the beta values of OxBS samples.

**Pyrosequencing.** Genomic DNA was isolated with the NucleoSpin Tissue kit (Macherey & Nagel, Düren, Germany) and bisulfite converted using the EZ DNA Methylation kit (Zymo Research, Irvine, CA, USA). Pyrosequencing was performed on a PyroMark ID System (Biotage, Uppsala, Sweden). Primers for pyrosequencing were designed with the PSQ assay design software (Biotage; Supplemental Table S3). DNAm levels were determined with the Pyro-Q-CpG Software (Biotage). To train the epigenetic predictor on pyrosequencing data we divided the pyrosequencing samples into a training and validation set (Supplemental Table S4 and Supplemental Data 2). The multivariable linear regression model based on DNAm levels (β-values) at the four CpGs in α = ALOX12 (cg03762994), β = DOK6 (cg25968937), γ = LTC4S (cg26683398) and δ = TNNI3K (cg05264232) was as follows:

$$\text{Predicted passage} = 39.0341 - -10.9266\alpha - 0.4219\beta + 5.8979\gamma - 38.889\delta \quad (1)$$

Finally, we used the R package caret[52] to perform 10-fold cross-validation on the training dataset.

**Lentiviral barcode-RGB marking.** Primary MSCs in umbilical cord tissue were transduced with three different lentiviral vectors containing the fluorescent proteins mCherry (red), Venus (green) or Cerulean (blue) and a barcode of 16 random and 15 vector-backbone-specific nucleotides, as previously published[30]. Clonal dynamics were assessed at each passage by flow cytometry using a BD LSR II flow cytometer (BD Biosciences, Heidelberg, Germany) and by deep sequencing of PCR amplified barcodes using ion torrent sequencing. Area plots were produced with the R package ggplot2. The Shannon-Index was calculated with the following formula:

$$H' = -\sum_i p_i {}^*\ln p_i \quad (2)$$

where $p_i$ is the proportion of distinct RGB clones or DNAm patterns, respectively.

**Barcoded-bisulfite-amplicon sequencing.** Bisulfite-converted DNA was used for a nested PCR using the PyroMark PCR kit (Qiagen; primers are provided in Supplemental Table S5). The second PCR added barcoded Illumina adapters that allowed to distinguish donors and passages, as described before[8]. Amplicons were pooled and sequenced on an Illumina MiSeq lane with the v2 nano reagents (Illumina) in 250 PE mode. Bisulfite converted sequencing data were analyzed using TrimGalore, Bismark[53] and bowtie2[54]. Mean sequencing coverage of amplicons was ~3900 reads per amplicon (Supplemental Data 3). Further pattern analysis and visualization was performed with custom perl and R scripts or with R package ggplot2 for area plots. Pearson correlation of neighboring CpGs and the corresponding heatmap were produced with the python packages scipy and seaborn, respectively. Single read predictions were performed as described before[31]. In short, single reads of the BBA-Seq amplicons were assigned to their most likely passage number (from 0 to 50) based on their binary sequel of methylated and

unmethylated CpG sites. Probabilities of passage numbers were based on linear regression models at each CpG site, retrieved from the training dataset. Finally, we calculated the mean passage number for each sample based on all sequencing reads. Further details on the rational and derivation of the mathematical model are provided in our previous work[31].

**Analysis of hemimethylation.** Hemimethylation analysis was modified from a protocol by Laird et al.[15]. Genomic DNA (4 µg, three donors in passages 2 and 13) was digested with restriction enzymes that cut close to our CpG of interest: AccI (CASR, TNNI3K, C12orf12), DdeI (KRTAP13.3, LTC4S) and CviQI (DOK6, GRM7, PRAMEF2, SELP, GNAS). Hairpin linkers (Supplemental Table S6) were denatured at 95 °C and subsequently folded by slow cooling to room temperature. Ligation was performed with 4000 U ligase and 3.3 µM hairpin linker DNA over night at 16 °C. The ligated DNA was denatured with 0.3 M NaOH at 42 °C for 15 min and 99 °C for 2 min before a 0.4 g/ml sodium bisulfite and 1 mg/ml hydroquinone solution was added. Bisulfite conversion was performed at 55 °C overnight with 10 intervening denaturation steps at 99 °C to prevent renaturation of hairpin structures. The regions of interest were subsequently amplified by PCR using the PyroMark PCR kit (Qiagen; primers listed in Supplemental Table S7). Illumina adapters were ligated using the GeneRead DNA Library I Core Kit (Qiagen) and GeneRead DNA I Amp Kit (Qiagen). Final library cleaning was performed with the Select-a-size DNA Clean & Concentrator kit (Zymo Research). MiSeq v2 nano reagents (Illumina) were used for library dilution to 4 nM and 20% PhiX were spiked in to increase sequencing diversity. Sequencing was performed on an Illumina MiSeq in 250 PE mode and analyzed as described above. Directional clustering of UMI tools was used to group the unique molecular identifiers[55]. Mean sequencing coverage of hairpin amplicons was ~9600 reads per amplicon (Supplemental Data 4).

**Circular chromatin conformation capture (4C).** Ten million cells from two MSC preparations at early (P2 and 3) and late (P7 and 9) passages were cross-linked with 4% paraformaldehyde (Electron Microscopy Sciences, Hatfield, PA, USA) for 10 min., harvested into ice cold PBS with 0.125 M glycine, and frozen under a protease inhibitor cocktail (Roche, Basel, Suisse). 4C-seq was performed as described before[56] using ApoI as the primary and DpnII as the secondary restriction enzyme. Bait-specific primers for the circularized inverse PCR are listed in Supplemental Table S8. Amplicons were sequenced on a HiSeq2500 platform (Illumina), mapped to the reference genome (hg19) and analyzed with the Hidden-Markov-Model-based tool 4Cker[57]. Numbers of sequenced reads per sample ranged from ~0.7 to 12 million reads with a mean sequenced read number of ~7 million reads per sample. 4Cker corrects for increasing signal noise in trans chromosomal interactions and far-cis chromosomal interactions by adaptive window sizes. We used the kth-next-neighbor adaptive window sizes of k = 5 for nearbait (10 MB around the bait region of interest) and cis interaction analysis and k = 20 for trans chromosomal interactions. We focused particularly on the high-interacting reads that were called in both replicates. Differential interactions were called with 4Cker, which uses DESeq2 and a FDR corrected p-value < 0.5 to call significant differential interactions. Circos-plots were generated with the R package RCircos[58]. For comparison we used DNAm profiles of MSCs at early (n = 5) and late passages (n = 5; GSE37067)[22]. CTCF and CTCFL enrichment in interacting regions was tested with the RGT motif enrichment tool (http://www.regulatory-genomics.org/motif-analysis/introduction). Motif enrichment was tested within high-interacting regions called by 4Cker ranging approximately from 1.1 to 336 kilobases and a mean region length of 23 kilobases within nearbait and from 1.3 to 950 kilobases with a mean region length of 86 kilobases within cis regions. Enrichment of interactions with LADs of human fibroblasts[32] were analyzed in comparison to randomly chosen regions of similar sizes by Fisher's exact test using R stats.

**Fluorescence microscopy of CTCF.** Staining of CTCF (Rabbit polyclonal anti-CTCF, AB_2614983; Active Motive) and counterstaining with DAPI was performed as described in detail before[20]. For image acquisition, a widefield LeicaDMI 6000B with a HCX PL APO 63x/1.40 (Oil) objective was used.

**CTCF ChIP-seq.** Chromatin Immunoprecipitation of CTCF was performed using the ChIP-IT PBMC kit with the fixation protocol of the ChIP-IT High Sensitivity kit (both Active Motif, Carlsbad, CA, USA). In brief, MSCs of early passage (P 2, n = 3) and late passage (P 8–14, n = 3) were cross-linked with 1.2% formaldehyde (AppliChem, Darmstadt, Germany) for 15 min at room temperature. After cell lysis, chromatin fragmentation was performed for 20 min (20 cycles a´ 30 s on/30 sec off, total on-time 10 min) with a Bioruptor® Pico Sonicator Device (Diagenode). 5–33 µg of sonicated chromatin was incubated with 4 µg ChIP-validated CTCF antibody (Active Motif) overnight on an end-to-end rotator at 4 °C. For immunoprecipitation, samples were incubated with protein G agarose beads (Active Motife) at 4 °C for 3 h. Finally, the cross-link was reversed, the DNA was proteinase K digested and purified.

Sequencing and library preparation were conducted by the IZKF Sequencing Core Facility of RWTH Aachen University Medical school. The samples were sequenced in paired-end mode on the Illumina NextSeq 500. The quality of the resulting reads was checked by FastQC (Available online at: http://www.bioinformatics.babraham.ac.uk/projects/fastqc) before reads were aligned to the

hg19 genome assembly using bowtie2[54]. Reads approximately exhibited a 97% concordant alignment rate in all samples and sequencing depth ranged from 38 to 170 million reads. We used MACS2[59] for peak calling on each sample with the default parameters against the input control. All the peaks were filtered by signal >50 and selected with the overlapping of potential CTCF binding motif, which was obtained by RGT motif analysis (http://www.regulatory-genomics.org/motif-analysis). Enrichment of interactions between 4C high-interacting regions and CTCF ChIP-seq peaks of early and late passages were analyzed in comparison to randomly chosen regions of similar sizes by Fisher's exact test using R stats. The normalized coverage of each sample was calculated by Deeptools[60] bamCoverage with normalization of Reads Per Kilobase per Million mapped reads (RPKM), these normalized coverages were used for down-stream analyses. The correlation heatmap across the samples was generated by Deeptools multiBigwigSummary. All the potential CTCF binding sites were evaluated by the normalized coverage profiles and shown in MA plot by comparing early passages with late passages. The lineplots were calculated by a sliding window (size = 200 bp with step = 100 bp) on the normalized read counts on the extended sites (length = 2000 bp for each direction) of either CTCF motifs or various sets of CpGs. Statistical analysis was performed against these random CpG sites using the Kruskal–Wallis test with post hoc Dunn's test and Benjamini & Hochberg adjustment.

**Statistics and reproducibility**. Statistical tests were performed as described in the corresponding methods sections. All experiments were performed for at least two independent cell donors and at least four culture-associated CpGs.

**Reporting summary**. Further information on research design is available in the Nature Research Reporting Summary linked to this article.

## Data availability
Raw data of DNA methylation and hydroxymethylation profiles generated in this study and raw data as well as processed data of CTCF ChIP-seq (Figs. 6) and 4C-seq (Fig. 5) analysis were submitted to Gene Expression Omnibus (GEO): GSE144196. The publicly available datasets of Fig. 1a are provided in Supplemental Table S1, pyrosequencing data of Fig. 1b and c are provided as Supplemental Table S4 and Supplemental Data 2. All other data are available from authors upon reasonable request.

## Code availability
Publicly available software packages were used for data analysis as described in the corresponding methods sections. Workflows of data analysis can be provided upon request.

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

## Acknowledgements

We would like to thank all donors for their valuable support of our research and Minseung Choi (University of Washington, Washington, USA) for his input to the analysis of hairpin BBA-Seq. This work was particularly supported by the Else Kröner-Fresenius-Stiftung (A.P. and W.W.: 2014_A193, www.ekfs.de), by the Deutsche Forschungsgemeinschaft (W.W.: WA 1706/8-1; WA 1706/12-1; A.P.: UoC Advancer Research grant of the DFG Excellence Initiative, M.R.: RO 5102/1-1, www.dfg.de), by the German Ministry of Education and Research (WW: VIP+, 03VP06120, www.bmbf.de), by the Interdisciplinary Center for Clinical Research within the faculty of Medicine at the RWTH Aachen University (W.W.: IZKF O3-3, www.medizin.rwth-aachen.de/cms/Medizin/Die-Fakultaet/Einrichtungen/~dgun/IZKF-Aachen/), and by CMMC core funding (AP, www.cmmc-uni-koeln.de).

## Author contributions

J.F. contributed to experimental design, analysis, and writing of the manuscript; T.G., L.B., and A.P. performed 4C experiments; M.M. performed CTCF ChIP-seq experiment and C.-C.K. analyzed the ChIP-seq data; A. Selich, M.R., and A. Schambach performed barcoded-RGB-vector experiments and M.N. supported the analysis; R.S. helped to establish hairpin BBA-Seq; E.F.R., C.G., and A.O. contributed to long-term cell culture and cellular characterization; M.B. helped to sequence data; B.R. and A.D.H. contributed important material. W.W. contributed to experimental design, data analysis, and writing of the manuscript.

## Funding

## Competing interests

The authors declare the following competing interests: W.W. is cofounder of Cygenia GmbH (www.cygenia.com), which can provide service for epigenetic analysis to other scientists. J.F. contributes to this company, too. All other authors do not have a conflict of interest to declare.
