## [Peer Review File · Communications Biology]

Reviewers' comments:

Reviewer #1 (Remarks to the Author):

Upon prolonged cell culture, DNA methylation (DNAm) reproducibly changes at defined sites in the human genome. Here, the authors compile experiments to gain insight in the cause and consequences of these changes, including changes upon cellular reprogramming (Fig. 2), variation between cells and neighboring sites (Fig. 3 and 4) and links with CTCF binding and 3D genome organization (Fig. 5 and 6).

The reproducible changes in DNAm during prolonged culture are relevant both for studies using in-vitro models and studies on aging cells, where analogous changes are observed. Some of the findings in the manuscript provide interesting new insights in this process and are therefore of interest to these fields.

Whereas the first part of the manuscript is of high quality and easily accessible, the parts on 3D genome organization and CTCF binding have considerable issues. Before the manuscript can be considered for publication, these need to be thoroughly addressed.

Issues about the 4C-seq and CTCF ChIP-seq analysis:

1. The authors have used 4Cker to identify 'high interacting regions'. A visual inspection of panel 5A shows that the regions identified by 4Cker are particularly enriched in regions where the 4C signal is low. The authors should confirm if the output of the 4Cker tool correctly identifies regions of increased interactions (and not the inverted), or rather if this problem stems from either incorrect figure complication or figure resolution issues.
2. The raw sequencing data for the 4C-seq analysis should be made available (e.g. at the GEO repository).
3. I have difficulties interpreting panels 5C-F. Whereas panels 5C and D indicate that the total number of interactions increase upon passage number, the number of significantly different interactions decrease (panels 5E and F). Does this mean that many shared interactions become less significant upon passage number? The interpretation of the figure could be helped by grouping panels together (i.e. C + E and D + F), for instance by using Venn-diagrams instead.
4. Panels 5H-J raise a number of questions/remarks:
 - These panels will be easier to access if the authors provide more explicit description of what the Pearson correlation represents (both in the text and figure legend).
 - Several mentions are made about preferred or non-preferred interactions. The authors should provide a statistical comparison between early and late passage interactions to help the interpretation of these data.
 - As mentioned in panel 5G, KRTAP13.3 interacts mostly with nearbait hypomethylated regions. According to panel 5H, the entire nearbait region becomes further hypomethylated (black line) upon increased passage, yet the DNAm state of KRTAP13.3 interacting regions does not change (red line). The authors should reflect how the 3D organization of this viewpoint appears mostly stable, despite located in a genome region that undergoes considerable reorganization of DNAm.
5. Panel 5K: the authors should more explicitly mention the origin of the LAD data, including that these data are not from the same cell type.
6. From line 269, the authors mention that only KRTAP13.3 significantly interacts with LADs. It is unclear though how this is determined, as the cis interactions of CASR and LTC4S overlap within a comparable range with LADs.
7. Line 308: the finding that hypermethylated CpGs are enriched in CTCF binding sites is unexpected, as CTCF binding is thought to be inhibited by DNAm (see e.g. references 41 and 42). The authors should reflect on this finding, particularly linked to the possibility that the high GC-content of the CTCF binding motif may introduce a bias to detect hypermethylated CpGs.
8. Line 316: figure 6 shows there is little difference in CTCF binding between early and late passage cells, so it's unclear to me what this conclusion is based on.

9. Lines 347: it should be explicitly mentioned here that the identified differences in 3D chromatin structure appear not to be linked to changes in DNAm.

Other remarks:

- In figure 2, it would be preferred to move the color legend for DNA methylation next to panels A and B. In figure S2, the DNAm legend is missing.
- Figure 3A and B are based on 10 CpGs, including the 4 sites identified in Figure 1. The remainder of figures 3 and S3 are based on half of these sites, revealing a considerable variation between sites: panel S3B (right) indicates that the clonality of LTC4C, DOK6 and ALOX12 remains mostly stable during prolonged culture. To allow for a better interpretation of the data, the authors should include all 10 loci in all the panels.

Reviewer #2 (Remarks to the Author):

In their manuscript "Epigenetic drift during long-term culture of cells in vitro" Franzen J. et al. ask whether highly reproducible DNA methylation changes at specific CpGs during cell culture expansion are directly regulated by a specific mechanism or caused by gradual deregulation of the epigenetic state. The work points out that it is so far unclear how these DNA methylation patterns evolve during culture expansion and why they occur at specific genomic regions.

Using interesting approaches and models the authors give some interesting new insights on the characteristics of these intriguing CpGs and elucidate how these epigenetic changes evolve over cell culture expansion. The manuscript concludes that during cell expansion stably modulated CpGs are the result of an indirect epigenetic drift rather than the result of a targeted epigenetic mechanism involving DNA methylases. One of the main arguments for these conclusions is that if the generation of these stably modulated CpGs would be related to active involvement of DNA methylases, then neighboring CpGs would be modified as well as seen when for example DNMTs are targeted to specific genomic locations by CRISPR technology. The notion that the generation of spatially highly specific CpG methylation sites which are stably transmitted over several cell divisions is the result of an indirect epigenetic drift seems still rather counterintuitive. In the case of CRISPR directed DNMTs or TETs to specific genomic locations, the variability/efficiency of CpG methylation/hydroxymethylation around the targeted regions rises with increasing distance to the target site as well. To prove the point made, it would be necessary to specifically target DNMTs by CRISPR/dCas9 to some of the sites containing CpGs exhibiting cell expansion related methylation changes to exclude that methylation specificity at these single CpGs are not due to some structural chromatin peculiarity at these locations.

Further, the data presented shows that highly reproducible DNA methylation changes during cell culture expansion represent both, hyper- and hypo methylated CpGs. This presupposes that for the hyper-methylated CpGs, methylation needs to be actively passed over the cell cycles by a DNMT and for the hypo-methylated CpGs demethylation might occur via passive DNA demethylation or going through the active DNA demethylation cycle. Using the same argument as above, the question is then on how these single CpGs get methylated/spared or selectively demethylated as writers or erasers tend to affect a broader region and not only specific CpGs. Figure S3D and Figure 4A show that DNA methylation patterns of neighboring CpG sites fluctuate over multiple passages. However, as shown in Figure 4B, the methylation status for several neighboring CpGs within the analyzed region of GRM7 shows very high correlations with passage number as well. In addition, as shown in Figure 4F, the methylation patterns between donors as well as between passage numbers show a certain degree of similarity. Therefore the question arises if not only the methylation status of the single CpG's but the methylation status or pattern at these specific regions needs to be considered. In the context that specific, highly reproducible and expansion dependent DNA methylation changes were found at single CpGs a few further questions might be considered: 1. Do CpGs carrying highly reproducible DNA methylation changes during cell expansion have a specific CpG density in their proximity? 2. Does the

occurrence of these CpGs somehow relate to the nucleosome size (1 CpG in/per 146bp)? Would these or other structural properties/particularities at these or around these specific locations explain the methylation change at the single CpG level and therefore still integrate a tightly controlled mechanism?

On a technical note, the 450K methylation BeadChIP used in this study covers mainly gene regions and CpG islands (thus mostly regions around genes, gene bodies and regions involved in the proximate control of gene expression) but fails to fully cover the genome-wide CpG landscape. Intergenic regions cover a substantial part of the genome and regulatory elements found in these regions like distant (super)enhancers play important roles in chromatin structure and nuclear organization. The authors were able to show nuclear organization changes between early and late passages, but lack of a full CpG coverage might cause to miss important culture expansion-dependent DNAm changes and potential structural links between the stably modulated CpGs and chromatin organization. Furthermore, the 450K methylation BeadChIP protocol uses bisulfite technology which does not discriminate between 5mC and 5hmC. The same is true for the pyrosequencing approach utilized in this study. Although 5mC is much more abundant than 5hmC, 5mC and 5hmC represent two opposite marks (one found in closed and the other found in open chromatin) and therefore discrimination between the two would be important. In addition, DNA methylation marks by DNMTs are set during DNA replication while 5hmC marks are generated independently of DNA replication. This is an important difference which needs to be taken into consideration as it might indicate the chromatin status at the specific location and give some hints about the background and mechanism on how these marks are established.

In regard to statistical evaluation as well as analysis tools and scripts, state of the art approaches were used and experimental details are coherently outlined.

Reviewer #3 (Remarks to the Author):

Major Comments:

Anyone studying epigenetic changes, specifically, researchers studying DNA methylation will find this study fascinating however there are a number of comments in the article that reduce my enthusiasm centred around the conclusions of the study which need to be clarified.

1. Culture associated DNA methylation changes is associated with stochastic epigenetic drift (pg14)
2. The above are independent of site-specific TF binding (pg14)

The conclusions are based on four culture-associated regions for DNA methylation and 4C and the analysis of 2000 randomly chosen CG sites from the Illumina 450k beadchip.

The data is interesting and supports the above concept, however the authors should be considerate of language used in their descriptions of culture associated DNA methylation changes which tend to be over generalised and should emphasise precisely the number of genomic sites/regions actually investigated which is four CG sites plus 6 previously described. Taken together, these 10 CG sites are important but do not reflect generalised stochastic epigenetic drift.

the authors will need to specifically describe the culture associated DNA methylation changes are specific to cell culture and cell type conditions and limited to genomic sites investigated and is not necessarily generalizable.

Limitations of the study are not discussed.

Minor Comments:

Abstract - issues of spelling

Reviewers

Communications Biology

Aachen, 26.02.2021

Dear Reviewers of Communications Biology,

We are glad that you found merit in our work and that you provided helpful and constructive comments. Each of your suggestions has been addressed below and the corresponding changes are highlighted in red. We have now reanalysed and reformatted the 4C data, included additional data to address potential 5hmC, and we have more critically discussed limitations of our methods.

We feel that the manuscript significantly improved by the revision and we are looking forward to your decision.

With best regards,

Wolfgang Wagner
On behalf of all coauthors

Comments of Reviewer 1:

Upon prolonged cell culture, DNA methylation (DNAm) reproducibly changes at defined sites in the human genome. Here, the authors compile experiments to gain insight in the cause and consequences of these changes, including changes upon cellular reprogramming (Fig. 2), variation between cells and neighboring sites (Fig. 3 and 4) and links with CTCF binding and 3D genome organization (Fig. 5 and 6).

The reproducible changes in DNAm during prolonged culture are relevant both for studies using in-vitro models and studies on aging cells, where analogous changes are observed. Some of the findings in the manuscript provide interesting new insights in this process and are therefore of interest to these fields.

Whereas the first part of the manuscript is of high quality and easily accessible, the parts on 3D genome organization and CTCF binding have considerable issues. Before the manuscript can be considered for publication, these need to be thoroughly addressed.

We thank the reviewer for this encouraging feedback. In fact, we agree that the part of the 3D genome organization and CTCF binding were not well presented. Following this advice, the data were reanalyzed and presented in a different way, as indicated below.

Issues about the 4C-seq and CTCF ChIP-seq analysis:

1. The authors have used 4Cker to identify 'high interacting regions'. A visual inspection of panel 5A shows that the regions identified by 4Cker are particularly enriched in regions where the 4C signal is low. The authors should confirm if the output of the 4Cker tool correctly identifies regions of increased interactions (and not the inverted), or rather if this problem stems from either incorrect figure complication or figure resolution issues.

Indeed, the high interacting regions called by 4Cker tool did not seem to correlate with the peaks. As expected by the reviewer this was due to the resolution. We have now exchanged **figure 5A** and zoomed further into the nearbait region (4MB instead of 10 MB). The regions identified by 4Cker are now clearly at sites with increased signals.

2. The raw sequencing data for the 4C-seq analysis should be made available (e.g. at the GEO repository).

We now uploaded the sequencing data of the 4C-seq experiment to the GEO repository and data will be available upon publication under GSE165603 in the SuperSeries GSE144196 (reviewer token: *mxmjomesndyftyn*).

3. I have difficulties interpreting panels 5C-F. Whereas panels 5C and D indicate that the total number of interactions increase upon passage number, the number of significantly different interactions decrease (panels 5E and F). Does this mean that many shared interactions become less significant upon passage number? The interpretation of the figure could be helped by grouping panels together (i.e. C + E and D + F), for instance by using Venn-diagrams instead.

With the 4C analysis, we particularly wanted to analyze if genomic regions with senescence-associated DNAm changes preferentially interact with other senescence-associated regions - and we agree that the previous presentation was rather distracting. The former panels 5C and D demonstrated the total numbers of high interacting regions called by 4cker in early and late passages. Panels 5E and F showed the number of interactions identified as significantly different between early and late passages. Due to signal heterogeneity between the two donors it might occur that some of the differential regions do not reach the significance level, although the regions are either only called in early or late passage. We feel that it would be difficult to depict these differences in Venn diagrams. However, since this aspect was not the main focus of this analysis, we have now

shifted the panels to the **supplementary figures S5A and B** and provided further explanations in the figure legends.

4. Panels 5H-J raise a number of questions/remarks:

- These panels will be easier to access if the authors provide more explicit description of what the Pearson correlation represents (both in the text and figure legend).

We have now omitted the plots with DNAm [%] (previous figure 5G) and reformatted the Pearson correlation plots (previous figures 5H-J) as violin plots and with controls for all CpGs of the corresponding region. Furthermore, we now better explained the Pearson correlation and its meaning both in the text (**lines 264-275**) and **legend of Figure 5C and D**.

- Several mentions are made about preferred or non-preferred interactions. The authors should provide a statistical comparison between early and late passage interactions to help the interpretation of these data.

We now better specified the difference between so called “high interacting sites” as called by 4Cker and the statistically significantly changing sites between early and late passages, which are calculated by 4Cker using DESeq2 and an FDR adjusted p-value < 0.05. (**lines 255-260**)

- As mentioned in panel 5G, KRTAP13.3 interacts mostly with nearbait hypomethylated regions. According to panel 5H, the entire nearbait region becomes further hypomethylated (black line) upon increased passage, yet the DNAm state of KRTAP13.3 interacting regions does not change (red line). The authors should reflect how the 3D organization of this viewpoint appears mostly stable, despite located in a genome region that undergoes considerable reorganization of DNAm.

Following this advice, we have now extensively reanalyzed the data. In fact, the previously depicted mean of the Pearson correlation with passage number in the nearbait region at the KRTAP13.3 site was calculated based on a region, which was smaller than the actual nearbait region. We now corrected this mistake and show that the interacting regions harbor CpGs with similar culture-association as the overall CpGs in the nearbait region. The comparison is now much easier to read with the violin plots of the entire nearbait regions.

5. Panel 5K: the authors should more explicitly mention the origin of the LAD data, including that these data are not from the same cell type.

We now better clarified in the text and in the figure legend, that this data is not from the same cell type (but at least MSCs and fibroblasts are related) (**lines 285-286, 560, and legend of Figure 5E**).

6. From line 269, the authors mention that only KRTAP13.3 significantly interacts with LADs. It is unclear though how this is determined, as the cis interactions of CASR and LTC4S overlap within a comparable range with LADs.

The previous presentation for LAD association was difficult to read because we combined nearbait and cis, and used a different color code. We have now reformatted the **figures 5E** and **supplemental figure 5E** to better depict that only the hypomethylated region of KRTAP13.3 was enriched in interactions with LADs, whereas such interactions were depleted for hypermethylated regions – as compared to random genomic sites of the same sizes as the interaction sites.

7. Line 308: the finding that hypermethylated CpGs are enriched in CTCF binding sites is unexpected, as CTCF binding is thought to be inhibited by DNAm (see e.g. references 41 and 42). The authors should reflect on this finding, particularly linked to the possibility that the high GC-content of the CTCF binding motif may introduce a bias to detect hypermethylated CpGs.

The reviewer raises an important point that the CG-content of the selected sequences may affect CTCF binding. However, this would affect hyper- as well as hypomethylated CpGs. Please note that hypermethylated CpGs are not necessarily highly methylated CpGs (they just become more methylated during culture expansion). In fact, Figure 2A demonstrates that hypermethylated CpGs are overall less methylated than hypomethylated CpGs, particularly at early passages, which is in line with the above mentioned finding that CTCF binding is thought to be inhibited by DNAm. This was now added to the discussion (lines 374-376).

8. Line 316: figure 6 shows there is little difference in CTCF binding between early and late passage cells, so it's unclear to me what this conclusion is based on.

As suggested, we have now removed the statement “These results indicate that the formation of CTCF associated chromatin loops is relevant for the stochastic gains and losses of DNAm during culture expansion.”

9. Lines 347: it should be explicitly mentioned here that the identified differences in 3D chromatin structure appear not to be linked to changes in DNAm.

As suggested, this is now mentioned in the discussion (line 367).

Other remarks:

- In figure 2, it would be preferred to move the color legend for DNA methylation next to panels A and B. In figure S2, the DNAm legend is missing.

We now moved the color legend for DNA methylation next to panel A and B of figure 2 and included the legend in figure S2.

- Figure 3A and B are based on 10 CpGs, including the 4 sites identified in Figure 1. The remainder of figures 3 and S3 are based on half of these sites, revealing a considerable variation between sites: panel S3B (right) indicates that the clonality of LTC4C, DOK6 and ALOX12 remains mostly stable during prolonged culture. To allow for a better interpretation of the data, the authors should include all 10 loci in all the panels.

Not all of the 10 amplicons comprise neighboring CpG sites. This is the reason why we analyzed the Shannon index and DNAm patterns only for five amplicons. Furthermore, we clarified that DNAm patterns overall remained stable or became even more diverse during culture expansion (lines 191-194).

Comments of Reviewer 2:

In their manuscript “Epigenetic drift during long-term culture of cells in vitro” Franzen J. et al. ask whether highly reproducible DNA methylation changes at specific CpGs during cell culture expansion are directly regulated by a specific mechanism or caused by gradual deregulation of the epigenetic state. The work points out that it is so far unclear how these DNA methylation patterns evolve during culture expansion and why they occur at specific genomic regions.

Using interesting approaches and models the authors give some interesting new insights on the characteristics of these intriguing CpGs and elucidate how these epigenetic changes evolve over cell culture expansion.

We would like to thank the reviewer for this positive feedback.

The manuscript concludes that during cell expansion stably modulated CpGs are the result of an indirect epigenetic drift rather than the result of a targeted epigenetic mechanism involving DNA methylases. One of the main arguments for these conclusions is that if the generation of these stably modulated CpGs would be related to active involvement of DNA methylases, then neighboring CpGs would be modified as well as seen when for example DNMTs are targeted to specific genomic locations by CRISPR technology. The notion that the generation of spatially highly specific CpG methylation sites which are stably transmitted over several cell divisions is the result of an indirect epigenetic drift seems still rather counterintuitive. In the case of CRISPR directed DNMTs or TETs to specific genomic locations, the variability/efficiency of CpG methylation/hydroxymethylation around the targeted regions rises with increasing distance to the target site as well. To prove the point made, it would be necessary to specifically target DNMTs by CRISPR/dCas9 to some of the sites containing CpGs exhibiting cell expansion related methylation changes to exclude that methylation specificity at these single CpGs are not due to some structural chromatin peculiarity at these locations.

We agree that targeting DNMTs by CRISPR/dCas9 to some of the senescence-associated CpGs might be interesting. However, to determine if these regions have some structural chromatin peculiarities that hamper coherent DNAm, it would be necessary to address multiple senescence and non-senescence associated CpGs. As already mentioned in the response by the editors, we hope for your understanding that these experiments are indeed not within the scope of the current study.

Further, the data presented shows that highly reproducible DNA methylation changes during cell culture expansion represent both, hyper- and hypo methylated CpGs. This presupposes that for the hyper-methylated CpGs, methylation needs to be actively passed over the cell cycles by a DNMT and for the hypo-methylated CpGs demethylation might occur via passive DNA demethylation or going through the active DNA demethylation cycle. Using the same argument as above, the question is then on how these single CpGs get methylated/spared or selectively demethylated as writers or erasers tend to affect a broader region and not only specific CpGs. Figure S3D and Figure 4A show that DNA methylation patterns of neighboring CpG sites fluctuate over multiple passages. However, as shown in Figure 4B, the methylation status for several neighboring CpGs within the analyzed region of GRM7 shows very high correlations with passage number as well. In addition, as shown in Figure 4F, the methylation patterns between donors as well as between passage numbers show a certain degree of similarity. Therefore the question arises if not only the methylation status of the single CpG's but the methylation status or pattern at these specific regions needs to be considered. In the context that specific, highly reproducible and expansion dependent DNA methylation changes were found at single CpGs a few further questions might be considered: 1. Do CpGs

carrying highly reproducible DNA methylation changes during cell expansion have a specific CpG density in their proximity?

We have already addressed this important point in our previous work (Franzen *et al.* 2017, Aging Cell): senescence-associated CpGs are rather depleted from CpG islands and their shore/shelf regions. Hyper- and hypomethylated CpGs are enriched in open sea, 3'UTR and intergenic regions. This is now mentioned in the discussion (lines 395-397).

2. Does the occurrence of these CpGs somehow relate to the nucleosome size (1 CpG in/per 146bp)? Would these or other structural properties/particularities at these or around these specific locations explain the methylation change at the single CpG level and therefore still integrate a tightly controlled mechanism?

The idea that senescence-associated regions might be somehow related to nucleosome size is interesting. Our BBA-seq analysis indicates that neighboring CpGs are also culture-associated. Thus, it is not regularly 1 CpG in/per 146bp. The results in Figure 4B might be compatible with the hypothesis that the differentially methylated regions are overall related to nucleosome size. However, to really address this issue a genome wide approach would be necessary (e.g. WGBS with enough read depths and samples of many passages). We now discuss the aspect (Lines 386-388).

On a technical note, the 450K methylation BeadChIP used in this study covers mainly gene regions and CpG islands (thus mostly regions around genes, gene bodies and regions involved in the proximate control of gene expression) but fails to fully cover the genome-wide CpG landscape. Intergenic regions cover a substantial part of the genome and regulatory elements found in these regions like distant (super)enhancers play important roles in chromatin structure and nuclear organization. The authors were able to show nuclear organization changes between early and late passages, but lack of a full CpG coverage might cause to miss important culture expansion-dependent DNAm changes and potential structural links between the stably modulated CpGs and chromatin organization.

As suggested, we now further discussed the limitations of the Illumina BeadChips in the text. We agree that for a correlation with chromatin organization or nucleosome positioning other methods, such as WGBS, would be better suited (but much more expensive for the required coverage and sample number; lines 386-388).

Furthermore, the 450K methylation BeadChIP protocol uses bisulfite technology which does not discriminate between 5mC and 5hmC. The same is true for the pyrosequencing approach utilized in this study. Although 5mC is much more abundant than 5hmC, 5mC and 5hmC represent two opposite marks (one found in closed and the other found in open chromatin) and therefore discrimination between the two would be important. In addition, DNA methylation marks by DNMTs are set during DNA replication while 5hmC marks are generated independently of DNA replication. This is an important difference which needs to be taken into consideration as it might indicate the chromatin status at the specific location and give some hints about the background and mechanism on how these marks are established.

We have now included additional data for oxBS-450K. This method uses oxidative bisulfite (oxBS) chemistry to specifically detect both 5mC and 5hmC with 450K BeadChips (Steward *et al.*, Methods, 2015; TrueMethyl® protocol). As indicated by the reviewer the average predicted 5hmC levels were indeed very low (about 0.3%) and there was no difference at the culture-associated regions. Thus, 5hmC does not seem to play a central role for culture-associated CpGs. The additional results are provided in lines 127-135 and figure S1E; additional data have been deposited at GEO (SuperSeries GSE144196; lines 127-135; Supplemental figure S1E).

In regard to statistical evaluation as well as analysis tools and scripts, state of the art approaches were used and experimental details are coherently outlined.

We would like to thank the reviewer for the supportive feedback.

Comments of Reviewer 3:

Major Comments:

Anyone studying epigenetic changes, specifically, researchers studying DNA methylation will find this study fascinating however there are a number of comments in the article that reduce my enthusiasm centred around the conclusions of the study which need to be clarified. 1. Culture associated DNA methylation changes is associated with stochastic epigenetic drift (pg14); 2. The above are independent of site-specific TF binding (pg14).

We would like to thank the reviewer for the overall positive feedback. In fact, we summarized several findings that may support the notion that culture associated DNAm changes are associated with epigenetic drift, but we did not investigate TF binding in this study. In our previous work, we demonstrated that transcription factor binding motifs (e.g. EGR1, TFAP2A, and ETS1) were significantly enriched in differentially methylated regions (Hänzelmann et al., Clinical Epigenetics, 2015). This is now better clarified in the discussion (Lines 380-383).

The conclusions are based on four culture-associated regions for DNA methylation and 4C and the analysis of 2000 randomly chosen CG sites from the Illumina 450k beadchip. The data is interesting and supports the above concept, however the authors should be considerate of language used in their descriptions of culture associated DNA methylation changes which tend to be over generalised and should emphasise precisely the number of genomic sites/regions actually investigated which is four CG sites plus 6 previously described. Taken together, these 10 CG sites are important but do not reflect generalised stochastic epigenetic drift. The authors will need to specifically describe the culture associated DNA methylation changes are specific to cell culture and cell type conditions and limited to genomic sites investigated and is not necessarily generalizable.

As suggested, we have now revised the wording throughout the manuscript and we have added a paragraph to the conclusions to clarify the limitations of our study (Lines 387-393).

Limitations of the study are not discussed.

As indicated above, we have now discussed limitations of the study, particularly in the conclusion.

Minor Comments:

Abstract - issues of spelling

We have corrected spelling mistakes in the abstract.

We want to thank all three reviewers for very valuable and constructive suggestions that helped to improve our manuscript.

REVIEWERS' COMMENTS:

Reviewer #1 (Remarks to the Author):

In this considerably reworked manuscript, the authors have sufficiently addressed the questions and remarks by the reviewers. As a result, the data is more correctly presented and better accessible. Based on these changes, I now support the publication of the manuscript in Communications Biology.

Reviewer #3 (Remarks to the Author):

The authors have addressed the comments and concerns.